# Potential of Organic Acids, Essential Oils and Their Blends in Pig Diets as Alternatives to Antibiotic Growth Promoters

**DOI:** 10.3390/ani14050762

**Published:** 2024-02-29

**Authors:** Rumbidzai Blessing Nhara, Upenyu Marume, Carlos Wyson Tawanda Nantapo

**Affiliations:** 1Department of Animal Sciences, School of Agriculture Science, Faculty of Natural and Agricultural Science, North-West University, P Bag X 2046, Mmabatho 2735, South Africa; nantapocarlos@gmail.com; 2Department of Livestock, Wildlife and Fisheries, Great Zimbabwe University, Masvingo P.O. Box 1235, Zimbabwe; 3Food Security and Safety Niche Area, Faculty of Natural and Agricultural Science, North-West University, P Bag X 2046, Mmabatho 2735, South Africa

**Keywords:** antibiotic growth promoters, essential oils, organic acids, pig nutrition, welfare, meat quality

## Abstract

**Simple Summary:**

Organic acids and essential oils have been shown to be effective alternatives to antibiotic growth promoters in pig production. Organic acids and essential oils have antibacterial, antiviral, and antioxidant properties. The article will focus on the effectiveness of organic acids, essential oils, and their blends in pig diets as alternative antibiotic growth promoters. Furthermore, the effects of organic acids, essential oils, and their blends on growth performance, oxidative stress, and meat quality are examined. Organic acids and essential oils, which have antimicrobial properties, can be used in place of antibiotic growth promoters. The use of organic acids and essential oils as growth promoters enhances pig welfare and aids in the fight against antimicrobial resistance.

**Abstract:**

Over the years, the use of management and feeding strategies to enhance pig productivity while minimizing the use of antibiotic growth promoters has grown. Antibiotic growth promoters have been widely used as feed additives to reduce diet-related stress and improve pig performance. However, increasing concern about the consequences of long-term and increased use of antibiotic growth promoters in animal production has led to a paradigm shift towards the use of natural organic alternatives such as plant essential oils and organic acids in pig nutrition to enhance growth. Antibiotic growth promoters endanger human health by allowing multidrug-resistant genes to be transferred horizontally from non-pathogenic to pathogenic bacteria, as well as directly between animals and humans. Scientific research shows that alternative growth promoters such as essential oils and organic acids appear to improve pigs’ ability to prevent pathogenic bacteria from colonizing the intestinal system, stabilizing the gut microflora and promoting eubiosis, as well as improving immunity and antioxidant stability. The purpose of this review was to provide an in-depth review of organic acids and essential oils as growth promoters in pig production, as well as their effects on productivity and meat quality. Organic acids and essential oils in pig diets are a safe way to improve pig performance and welfare while producing antibiotic-free pork.

## 1. Introduction

Over the years, there has been an increasing global interest in the development of management and feeding strategies that maximize pig productivity while minimizing the use of antibiotic growth promoters [1]. Antibiotic growth promoters were first used as feed additives to prevent the stress resulting from changing feed in the diet [2]. However, growing concern about the long-term and increased use of antibiotic growth promoters in animal production has resulted in a shift toward using natural organic alternatives to boost pig growth, such as plant essential oils and organic acids [3]. Antibiotic growth promoters are harmful to human health because multidrug-resistant genes can be transferred horizontally from non-pathogenic to pathogenic bacteria, resulting in the direct transfer of antibacterial-resistant bacteria from animals to humans [4]. Moreover, antibiotic growth promoters are not eco-friendly as their residues are found in soils and water, negatively impacting the ecosystem and its functions [5]. Alternative growth promoters such as essential oils and organic acids have been reported to improve a pig’s ability to prevent pathogenic bacteria from colonizing the intestinal system [2], stabilize the gut microflora, and promote eubiosis [6]. They also improve mineral utilization, act as an energy source, promote endogenous enzyme secretion, and improve immunity and antioxidant stability [2]. As a result, improved pig performance is comparable to that of antibiotic growth promoters [6,7,8]. The focus of this review is to provide a comprehensive account of organic acids and essential oils as growth promoters in pig production and how they impact productivity and meat quality in pigs. It will provide an overview of the mode of action, performance responses, and potential of essential oils and organic acids in the pig industry. 

## 2. Material and Methods

A systematic literature review was conducted using Web of Science, Google Scholar, Scopus, and PubMed to gather peer-reviewed papers from 1990 to 2023. The search criteria focused on antibiotic growth promoters, alternative growth promoters, essential oils, organic acids, pig nutrition, immune system, pig welfare, and pork quality. The data were analyzed to determine if essential oils and organic acids can be used as alternative antibiotic growth promoters in pig diets and to learn how their inclusion affects pig performance and pork quality. Data were analyzed, synthesized, and presented based on the key questions raised in the development of a review.

## 3. Mode of Action of Antibiotic Growth Promoters

A large proportion of the pigs produced in the world received antimicrobials in their feeds to counter post-weaning challenges. This equates to 70–80% of all pig starters, 70–80% of grower diets, and 50–60% of finisher feeds in Europe for the last few years [9]. Weaning piglets causes altered stomach pH, post-weaning diarrhea, and performance issues due to a lack of hydrochloric acid, which activates digestive enzymes. Insufficient hydrochloric acid and other environmental stressors disturb intestinal flora balance leading to a proliferation of pathogenic coliforms [10]. Antibiotics enhance feed conversion but do not impact carcass quality [11]. Antibiotics used in swine production can suppress or inhibit the growth of certain microorganisms [9]; however, their chemical composition and bacterial spectrum of antimicrobials vary widely. Antibiotics induce bacterial cell death by inhibiting essential cellular functions. Antibiotics can be classified based on the cellular component or system effect that may induce cell death, or merely inhibit cell growth [11]. Antibiotics can suppress the growth of pathogenic microbes by reducing competition for nutrients, hence reducing microbial metabolites that affect growth rate [11]. 

However, administering antibiotics to livestock has resulted in the problem of antimicrobial resistance. Antimicrobial resistance compromises the efficacy of preventing and treating a growing number of microbial infections. It arises as a result of natural selection and mutations resulting in antibiotics being ineffective, giving a survival advantage to the mutated strain [12]. Additionally, antibiotics used in livestock production are not fully absorbed and metabolized in animals, resulting in a large dose being highly active when excreted, causing the enrichment of antibiotic-resistant genes and a huge risk to the environment [13].

## 4. Organic Acids

### 4.1. Characteristics of Organics Acids

Organic acids are classified as any organic carboxylic acid, with or without keto, hydroxyl, or others from the non-amino functional group, including some short-chain fatty acids, but not all amino acids with the general R-COOH structure [14]. Organic acids are categorized into three main functional classes: short-chain fatty acids, medium-chain fatty acids, and tricarboxylic fatty acids. Short-chain fatty acids (SCFA), or simple mono-carboxylic acids (maximum 5 carbon atoms) such as acetic, formic, propionic, and butyric acids, are organic acids that are synthesized in the lower intestine by the microbial fermentation of indigestible sugars and amino acids. Medium-chain fatty acids (MCFA), or carboxylic acids containing a hydroxyl group (6–12C) such as malic, citric, tartaric, and lactic acids, represent an important energy source with higher antimicrobial activity due to their higher pKa. Lastly, tricarboxylic acids (TCA), or simply carboxylic acids with double bonds such as sorbic and fumaric acids, are intermediates in the Krebs cycle, and are involved in energy metabolism [14,15,16]. Other organic acids, such as sorbic, benzoic, and lactic acids, follow different structures. Organic acids are widely distributed in nature as normal constituents of plants and animal tissues, produced either by chemical synthesis or microbial fermentation of carbohydrates in the large intestine [17]. The dissociation constant (pKa) and carbon chain length (C1–C7) of common organic acids determine their antimicrobial efficacy. The sodium, potassium, and calcium salts of these acids, such as sodium benzoate, calcium formate, and calcium propionate, also have antimicrobial properties. Organic acids can be applied directly to feeds in solid or sprayed form, and are classified as feed preservatives or acidifiers. The efficacy of dietary organic acids depends largely on animal species, chemical composition (acid, salt), molecular weight, MIC value of the acid, targeted microbe species, gastrointestinal tract site, and buffering capacity of the feed [17,18]. Safety, odor, taste, and solubility in water are aspects to consider when applying organic acids to animal nutrition. Organic acids are weak acids that partly dissociate, and upon entering the bacterial cell membrane, they detach themselves in the inner, more alkaline part. The undissociated part then reduces the pH in the cytoplasmic area, disrupting the normal metabolic processes of certain types of bacteria, including *E. coli.*, *Listeria* spp., *Salmonella* spp., *Clostridia* spp., and some coliforms, thereby killing the cell [19]. In European markets, the demand for feed acidifiers grew by 6.6% from 2006–2012. The global market for feed acidifiers is projected to increase, with high demand in developing economies as well as demand for safe meat products from developed economies and an increasing world population [6]. The global market size for animal nutrition organic acids in 2020 was estimated at US$113.3 million, and is expected to expand by a 6.5% compound annual growth rate (CAGR) in 2021 to 187.1 million by 2028 [20]. As stated in this report, growth is driven by demand for low-cost renewable energy sources, as well as an increased need to replace traditional growth promoters. Furthermore, pigs and poultry were in high demand, with lactic acid accounting for 49.0% of total revenue. Table 1 shows the common organic acids used as dietary acidifiers for pigs and poultry.

### 4.2. Mode of Action of Organic Acids

The mode of action of common OAs is not yet fully understood. However, their mode of action may be partially due to factors such as, (a) inhibition of the development of pathogenic microbes in the gastrointestinal tract by reducing gut pH, (b) reduction of gastric emptying rates and maintenance of endogenous enzyme secretion, (c) mineral chelation and stimulation on intermediary metabolism, and (d) facilitation of proper digestion due to lower gastric pH and enhanced pepsin secretion [16].

### 4.3. Bactericidal Properties

Organic acids are weak acids, and in their undissociated state they can easily diffuse across cell membranes. In the cytoplasm, they dissociate and release hydrogen (H^+^) ions, which increases intracellular acidity of the cell, influencing cell metabolism and disrupting the normal microbial cell functioning [14]. Bacterial cells are forced to expend energy to expel the protons, leading to intracellular accumulation of RCOO^−^ acid anion. Accumulation of anions will interfere with RNA and DNA synthesis, resulting in impaired cell growth and multiplication as well as osmotic cell pressure, inducing both bactericidal and bacteriostatic effects [15]. The acid regulation properties of OAs allow them to reduce activity of harmful bacteria by altering the ambient pH value in the bacterial cell [6]. Organic acids have a stronger effect on the inhibition of gram-positive bacteria than gram negative bacteria due to structural differences in the cell membrane [19]. The efficiency of organic acids in reducing the microbial count is affected by the type of acid, temperature, buffering capacity, and water activity. Table 2 shows some common organic acids used in pig production. 

### 4.4. Lowering Stomach pH and Endogenous Enzyme Secretion

Short-chain fatty acids have a stimulating effect on both the endocrine and exocrine pancreatic secretions. Organic acids, when ingested, can create an acidic environment [2]. Low stomach pH alters gut microflora by reducing the non-acid tolerant bacterial species such as *E. coli* and *Salmonella* [14]. The acidic environment in the stomach activates the conversion of enzyme precursor pepsinogen to pepsin, which is responsible for protein digestion [1,32]. Organic acids elevate serum secretin content, stimulating pancreatic exocrine secretions and resulting in improved nutrient digestibility in the duodenum [33]. 

### 4.5. Energy Source and Mineral Utilization

Organic acids act as an energy source in the gut, as they are intermediary products of the tricarboxylic acid (TCA) cycle [34]. Their inclusion in diets helps in preventing tissue breakdown from gluconeogenesis and lipolysis [6,19]. Organic acid anions can form complexes with minerals like calcium, phosphorous, magnesium, and zinc, enhancing mineral digestion and reducing the excretion of supplemental minerals and nitrogen [2]. Organic acids can improve P solubility and phytate P utilization by competitively chelating Ca^2+^, reducing the formation of insoluble Ca phytate complexes [14]. Figure 1 illustrates the mode of action for organic acids when they are included in pig diets.

## 5. Essential Oils

### 5.1. Characteristics of Essential Oils

Essential oils are a mixture of various compounds, mainly terpenes and terpene derivatives. They are concentrated hydrophobic liquids containing volatile aromatic compounds produced by plants, stored in cavities, secretory cells, and epidermal cells. They are produced as secondary metabolites and they have antibacterial, antifungal, and antiviral properties [35]. These provide essential oils with the ability to replace antibiotic growth promoters and improve animal performance and health [26]. Ecological factors, species, climatic conditions, harvest time, the part of the plant used, and the method of isolation affect the chemical composition of essential oils and their efficacy [36]. Table 3 shows the commercial and non-commercial application of essential oils in pig nutrition and health.

### 5.2. Antibacterial 

Essential oils exhibit a wide spectrum of in vitro antibacterial activities against gram-negative and gram-positive bacteria including *E. coli*, *Salmonella*, *Staphylococcus*, *Klebsiella*, *Proteus*, *Bacillus*, and *Clostridium* species. Plant extracts kill pathogens due to their hydrophobicity and a high percentage of phenolic compounds. Bioactive compounds in essential oils prevent the development of virulent structures in bacteria, and active compounds disturb the enzyme system of bacteria blocking the virulence of the microbe [2]. Hydrophobicity properties of essential oils enable them to separate lipids present in the cell membrane of bacteria and mitochondria, making it more permeable and disturbing the cell structure [34]. This leads to cell death, due to the leakage of critical molecules and ions from the bacteria [41]. Essential oil containing phenolic groups exhibit antimicrobial properties through their delocalized electrons and the presence of a hydroxyl group on the phenolic ring. The oils initiate damage to bacterial cell membranes by compromising the pH homeostasis of the bacterial cell membranes [41]. Essential oils have a certain degree of selectivity towards gram-negative bacteria than gram-positive bacteria [1].

### 5.3. Antioxidant and Anti-Inflammatory Ability

The presence of phenolic OH and other pKa in essential oils contributes to their antioxidant properties. They act as hydrogen donors to the peroxy radical produced during lipid oxidation, inhibiting hydroxyl peroxide formation [26]. Essential oils improve redox balance in various organs and protect against oxidative damage caused by psychological stressors [42]. They also improve the oxidative capacity of meat, which influences its quality. Essential oils inhibit the production of pro-inflammatory cytokines and chemokines by endotoxin-stimulated immune and epithelial cells. Anti-inflammatory properties are partially mediated by inhibiting the NF-kB activation pathway [26], which prevents gut morphological changes, mucosa damage, increased mucosal permeability, impaired gut development, and poor nutrient absorption capacity [42].

### 5.4. Immune Stimulation 

Essential oils have immune-stimulating effects on the gastrointestinal tract commonly referred to as the gut-associated lymphoid tissue (GALT) (Figure 2). The gastrointestinal tract possesses the largest mass of lymphoid tissue and plays an important role in antigen defense in the body [32]. Supplementing essential oils improves the immune status of animals by increasing lymphocyte proliferation rate, phagocytosis rate, IgG, IgA, and IgM concentrations, as well as changes in lymphocyte distribution in the gut [9]. Table 4 indicates secondary metabolites found in essential oils.

**Table 4 animals-14-00762-t004:** Secondary metabolites in essential oils.

Compound Name	Classification	References
α-amyrin	Pentacyclic triterpene	[43]
1α,4α-dihydroxybishopsolicepolide	Guaianolide sesquiterpene lactone	[44]
12α,4α-dihydroxybishopsolicepolide	Sequiterpene	[43]
3,5-dicaffeoyl quinic acid	Phenylpropanoid	[45]
Acacetin	Flavone	[43]
Betulinic acid	Pentacyclic triterpenoid	[43]
Caffeic acid	Phenylpropanoid	[45]
Chlorogenic acid	Phenylpropanoid	[45]
Isoalantolactone	Sesquiterpene lactone	[46]
Phytol	Diterpene	[43]
Scopoletin	Coumarin	[43]
Yomogiartemin	Guaianolide sesquiterpene lactone	[44]

**Figure 2 animals-14-00762-f002:**
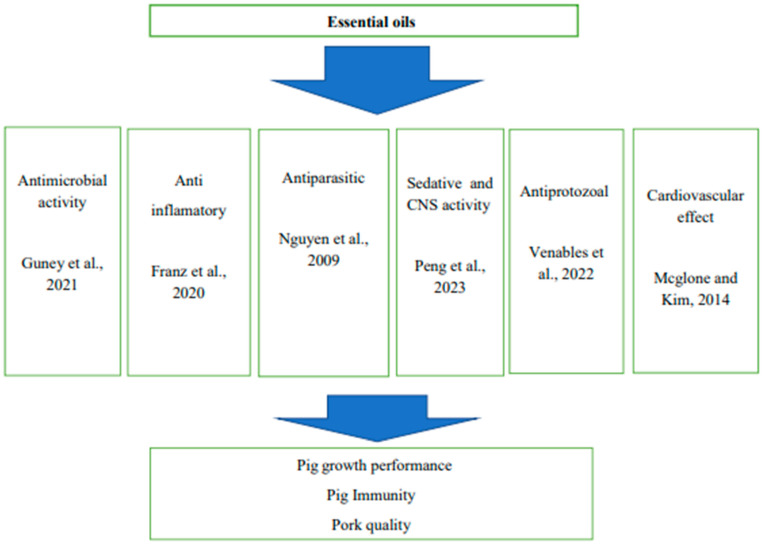
Summary of biological activities of essential oils [45,46,47,48,49,50].

## 6. Effect of Organic Acids and Essential Oils on Pig Performance

### 6.1. Influence on Voluntary Feed Intake (VFI)

Voluntary feed intake in pigs can be influenced by many factors, including dietary characteristics. Essential oils possess an intense smell which makes feed appealing, and pigs tend to consume feed more frequently and/or a larger amount at each meal before gut fill [4]. However, some essential oils result in reduced feed intake which can be attributed to an irritating smell that renders palatability displeasing [45]. Feed intake increased relative to control diets due to supplementation of essential oils ranged from 9% to 12% [49]. Lan, et al. [36] reported a range of 3–19% increase in feed intake. In weaned pigs’ supplementation of organic acids, essential oils and their blend showed to improve average daily feed intake [26]. Essential oils can increase feed palatability and intake due to their flavor-enhancing properties and odor. An increase in palatability associated with the supplementation of essential oils can be attributed to their antioxidative properties that help preserve feed quality and prevent the formation of unpleasant odors [42]. Some organic acids show no effect on feed intake as their performance effect in pigs is mainly on the gastrointestinal tract and nutrient metabolism [6]. Table 5 shows the effect of organic acids and essential oils on voluntary feed intake.

### 6.2. Influence on Nutrient Digestibility and Growth Efficiency

Studies on organic acids and essential oils have shown that they are beneficial and improve nutrient digestion. Formic acid and its salts lower the pH of the GIT, which increases the activity of digestive enzymes [34]. Organic acids have been shown to improve protein digestion by as much as 4%. Formic acid and its salts increased protein apparent tract digestibility, but did not improve ileal amino acid digestion (this could be due to diet acidification [46]). Citric acid improved the apparent total tract digestibility of protein, calcium, and phosphorous in sows [30]. Dietary benzoic acid improved the apparent digestibility of calcium and phosphorus in growing pigs, as well as crude protein in weanlings. Sows fed benzoic acid diets also had a high digestibility coefficient for organic matter, ether extract, crude protein, and crude fiber [52]. Dietary supplementation with protected acid blends increased the digestibility of dry matter, nitrogen, and energy in lactating sows [53]. Essential oils improved the apparent digestibility of crude protein and dry matter in swine. Studies have shown that phytogenic compounds can regulate ileal mucus gene expression and stimulate digestive secretions, thereby improving nutrient digestibility [42]. Essential oils act as a digestive stimulant by activating three (3) peripheral sensing mechanisms, known as oronasal. Oronasal sensing prepares the GI tract for food reception while also stimulating digestive secretion and gut motility [54].

### 6.3. Effect on Fecal Characteristics and Noxious Gas Production

Organic acids and essential oils can lower diarrheal incidence due to their ability to alter gut pH and microflora [2,6]. *Escherichia coli* is a major factor in causing infection and diarrhea in weaned pigs, and it affects growth performance [55]. Organic acids have been shown to increase the number of beneficial bacteria in the GIT in pigs and reduce the concentration of *E. coli* in feces. OAs penetrate bacterial cells in a non-dissociated form and disrupt the normal physiology of certain bacteria [14]. Organic acids and EO blends work in synergy to suppress growth of pathogenic microbes and promote the growth of beneficial microbes [55]. Formic acid, fumaric acid, and citric acid reduced the incidence and severity of diarrhea, increased microbial diversity in the GIT, and reduced *E. coli* counts while increasing lactobacilli counts [22,56,57]. Herb and plant extracts reduced the *E. coli* count and improved energy digestion [58,59]. Improved nutrient digestibility in pigs due to essential oils and organic acid supplementation had an impact on fecal noxious gas production. Kiarie et al. [60] state that protective acids reduced fecal emission of ammonia and hydrogen sulfide in lactating sows. The inclusion of humic substances in pig diets reduced ammonia emission by 3 to 18% in pig manure. Reduced aerial ammonia concentrations have beneficial effects on human health [52].

### 6.4. Effects on Gut Morphology and Gut Microflora

Low gastric pH due to the addition of acidifiers in diets maximizes the growth of beneficial bacteria in the GIT [61]. Organic acids and medium-chain fatty acids have been demonstrated to reduce pathogenic activity in pigs when fed in combinations rather than individually. They reduce the expression of pro-inflammatory cytokines and increase the proliferation of *Lactobacillus* bacteria. Formic acid added to diets of weaned pigs showed an increase in intestinal microbial diversity and a change in the concentration of certain microbes. A blend of organic acids also increased fecal *Lactobacillus* species and decreased *E. coli* fecal counts [62]. Plant extracts fed to pigs indicated improved gut health by modulating gut microbiota. Supplementation of essential oils decreased ileal total microbial mass and increased the lactobacilli to enterobacterial ratio. In vivo studies showed that essential oils increased the lactobacilli group and decreased *E. coli* and total coliform in piglets [42]. 

Organic acid and essential oil blends can increase the villous height of the duodenum. Essential oil supplements increased the villous height of the jejunum and the villous height to crypt ratio [26]. Supplements of essential oils reduced the number of intra epithelial and increased the villus height to crypt depth in the distal small intestines [2]. Essential oils decrease the number of pathogenic bacteria in the gut, favoring an increase in villus length, gut surface area, and crypt depth in the jejunum and colon [32]. Blends of medium-chain fatty acids and short-chain organic acids can be utilized by enterocytes as energy sources and attenuate the negative effect of weaning on villus length and crypt depth in pigs [34]. In weaner pigs, benzoic acid showed an increase in the villus height to crypt depth ratio [63]. However, in a study by Kong et al. [64], butyric acid did not affect the histology of grower-finishing pigs; only their mucosal depth was larger, and this can be attributed to better gut integrity in older animals.

### 6.5. Effect on Immune Status and Oxidative Stress

Supplementing pig diets with organic acid and essential oil has an impact on the immune system and the regulation of oxidative stress. Studies showed that essential oils reduced the numbers of intra epithelium lymphocytes in the mesenteric lymph nodes. Essential oils improve immunity and reduce the need for immune defense activity in the gut [32]. A mixture of carvacrol, cinnamaldehyde, and capsicum oleoresin decreased the population of intra epithelium lymphocytes in the jejunum and ileum of pigs [26]. Supplementation with butyric acid and essential oil reduced the white blood cell counts in growing pigs [62]. Li et al. [65] also stated that organic acid and essential oil blends can reduce the total white blood cell and neutrophil counts during the post-weaning period. An increase in WBC counts indicates systematic inflammation and the risk of bacterial infection [66].

Oxidative stress is commenced when the amount of ROS produced exceeds the neutralization ability of antioxidants. Excessive ROS leads to oxidative damage of proteins, lipids, and DNA, thereby destroying cell function [67]. Oxidative stress and inflammation are correlated physiological processes [68]. To prevent the accumulation of free radicals, cells develop defense mechanisms including the antioxidant enzymes and non-enzymatic antioxidants. Superoxide dismutase (SOD), glutathione peroxidase (GSH P_x_), and catalases (CAT) constitute antioxidant enzymes, whereas ascorbic acid, α tocopherol, Glutathione (GSH), carotenoids, and flavonoids are part of non-enzymatic antioxidants [69]. Oxidative stress represents an important chemical mechanism that leads to biological damage. Exposure of pigs to varied stressors leads to the increased production of ROS and the overwhelming of the antioxidant system. Oxidative stress is associated with reduced performance, decreased feed intake, diarrhea, and destruction of liver tissues [70]. In vivo experiments in pigs have shown that the antioxidant effects of essential oils reduce oxidative stress. Frankic et al. [69] state that carvacrol added in drinking water reduced the level of DNA lesions induced in freshly isolated hepatocytes and testicular cells in pigs. Mounir et al. [71] demonstrated that supplementation of plant extracts in pigs reduced DNA damage in lymphocytes which can be a potential benefit to the immune system. Organic acid Na-butyrate supplementation in gestating sow diets and pre-weanling diets had a positive effect on muscle and adipose tissue oxidative genes [70,71]. Improved antioxidant indices can prevent villi from radical-induced damage, which is correlated with better intestinal morphology and nutrient digestibility [72].

### 6.6. Effect on Growth Performance and Carcass Characteristics

Organic acids and essential oils improve the productivity of pigs to levels comparable to antibiotic growth promoters. According to Lückstädt et al. [6], organic acids improved daily weight gain, feed conversion rate, birth weight, weaning weight, and back fat thickness in pigs. The addition of fulvic acid in pig diets improved average daily gain and growth. Fulvic acid can improve the metabolism of proteins and carbohydrates [73]. A blend of organic acids showed an improvement in growth performance in older pigs and newly weaned pigs [74,75]. In the grower-finisher period, application of different levels and different sources of plant extracts showed positive effects on the growth performance of pigs [76]. [77] reported a higher average daily gain and feed conversion ratio in pigs fed garlic-treated diets. Bedford and Gong [78] observed a significant improvement in average daily gain and feed conversion ratio with the use of an herb mixture in pig diets from 25 to 105 kg.

### 6.7. Effect on Physicochemical Meat Properties

Meat quality characteristics are generally not affected by changes in diet composition, but can be influenced when diet alters carcass composition. According to Peng et al. [79], supplementing organic acids had no effect on marbling, meat color, cooking losses, drip losses, and water holding capacity in finisher pigs. Organic acids are considered growth promoters. However, there are no scientific studies confirming that they do not alter carcass composition. The addition of an organic acid and essential oil blend in grower-finisher pigs did not affect the pork’s ultimate pH, cooking loss, and shear force. Essential oils showed an effect on meat color by improving the oxidative stability of meat [12]. Rosemary essential oils reduced indicators of lipid oxidation and protein oxidation in pork [8]. Oregano essential oils when supplemented in pig diets prevented lipid oxidation but did not affect the cooking loss, drip loss, shear force, and chemical composition of pork [80]. Supplementing pigs with butyrate showed an effect on boar taint impacting pork sensory attributes. Butyrate has a regulatory effect on cell apoptosis and accumulation of androsterone in pigs, which causes boar taint [81]. However, there is a need for more research on the sensory effects of organic acids and essential oils in pork. Table 6 shows the effects of organic acids and essential oils on pig performance in relation to age.

### 6.8. Potential of Organic Acids and Essential Oils as Feed Additives in the Pig Industry

Figure 3 summarizes the potential benefits of organic acids, essential oils, and their blend in pig diets. The application of organic acidifiers and essential oils in pig diets has great potential in improving pig performance, pork quality, and reducing environmental pollution. This will, in turn, assist in meeting the ever-increasing demand for animal protein, positively affecting food and nutritional security. The use of organic acid and essential oils has the potential to reduce noxious gas emissions from pig manure, impacting climate change mitigation. The overall adoption of organic acids and essential oils in pig nutrition will lead to a drastic shift in the provision of safe pork that is antibiotic-free [2].

## 7. Conclusions

Organic acid and essential oils can improve nutrient digestibility, growth performance, carcass traits, gut morphology, microflora, meat quality, and chemical composition in pigs. The potential of organic acids and essential oils to improve pig performance and pork quality is comparable to that of antibiotic growth promoters and can be an alternative in smart pig production practices and the production of safe meat. However, information on their specific mode of action in growing pigs is still lacking, and there is a need for further research. Future studies are recommended on the effects of organic acid and essential oils on fermentation indices, immune and enzyme gene expression, fatty acid profile, and lipid quality indices.

## Figures and Tables

**Figure 1 animals-14-00762-f001:**
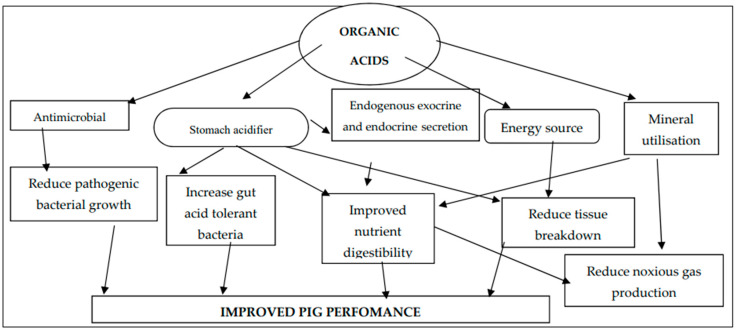
Summary of organic acidifier mode of action in pigs.

**Figure 3 animals-14-00762-f003:**
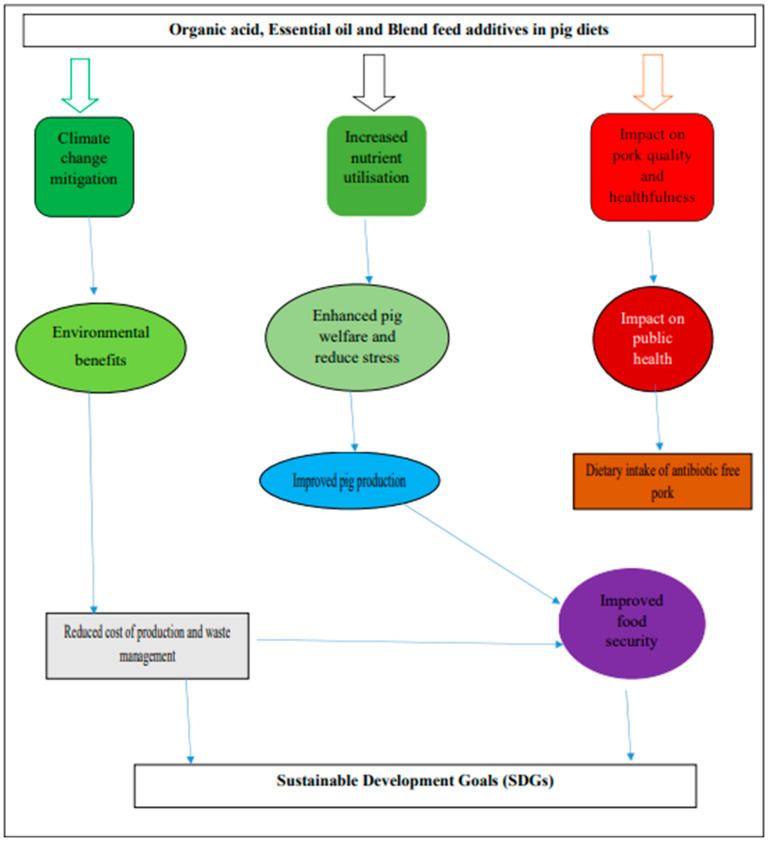
Summary on potential of organic acids and essential oil feed additives.

**Table 1 animals-14-00762-t001:** Chemical Properties of common organic acids used in animal nutrition.

OA	Chemical Name	Dissociation Constant, kP_a_	Physical Form
Tartaric	2,3-Dihydroxybutanedioic acid	2.93/4.23	Liquid
Formic	Methanoic Acid	3.75	Colorless liquid
Acetic	Ethanoic Acid	4.76	Colorless liquid
Propionic	2-Propanoic Acid	4.88	Colorless oily liquid
Caprylic acid	1-Octanoic acid	4.89	Colorless to light-yellow oily liquid
Butyric	Butanoic acid	4.82	Colorless oily liquid
Lactic	2-Hydroxypropanoic Acid	3.08	Colorless to yellow viscous liquid
Sorbic	2,4-Hexandienoic Acid	4.76	White crystalline powder or granules
Fumaric	2-Butenedioic Acid	3.02	White crystalline powder
Benzoic	Benzenecarboxylic acid	4.20	Colorless crystalline powder
Malic	Hydroxybutanedioic Acid	3.40/5.1	Liquid
Citric	2-Hydroxy-1,2,3- Propanetricarboxylic Acid	3.13/5.95/6.39	White or crystalline powder

Source [15,16,17,21].

**Table 2 animals-14-00762-t002:** Common organic acids used in pig production.

Organic Acid	Dietary Dose	Observations	References
Formic	1.4 g/kg	Positive auxinic effects, improved ADG, ADFI, and FCR during initial post-weaning 3 week period	[22]
Formic	6.4 g/kg	Higher microbiota diversity	[22]
Acid blend 1 Acid blend 2	231 FO, 124 AA, 127 LA, 133 PA and, 76 g kg^−1^ of CA. 50% acid (290 FO, 170 AA; 160 PA; 85 g kg^−1^ CA) + 50% LA on silica (517 LA; 7 FO and 20 g kg^−1^ AA)	Lower fecal coliform counts, inhibit early ileal microbiota development for all acids Acid 2 improved ADG in week 2	[23]
Malic	1.5% in weanling pigs	No improvement on growth performance	[24]
Benzoic	0.5% in nursery pigs	Inhibit pathogenic microbes maintain intestinal microecological balance, improve growth performance, and protein digestibility	[25]
Provenic	BA 50%, CF 3%, and FA 1%	Improved apparent total tract digestibility, fecal score, intestinal microbiota, and volatile fatty acid	[26]
Carbadox (PA and LA)	50 mg/kg	Reduced diarrhea scores	[27]
Orgacids^tm^ (FO, PA, LA, MA, TA, and CA)	2 kg/tonne	Low fecal pH and Enterobacteriaceae counts, higher *Lactobacillus* spp. counts, low meat cholesterol	[28]
Benzoic	5 g/kg	Improved BWG, ADFI, FCE	[29]
Citric	4 mmol/L	Improved immune function, reduced enterotoxigenic *E. coli* induced damage to the intestinal barrier of weaned piglets	[13]
Citric	5, 10, or 15 g/kg, during gestation and lactation	Increased total tract apparent digestibility of Cp and P Enhanced plasma and colostrum IgG and IgA Improved total protein of milk and colostrum	[30]
Matrix coated OA blend	0.2% in growing pigs	Enhanced growth performance and improved gut microbial population with no adverse effect on nutrient digestibility	[31]

PA—phosphoric acid; LA—lactic acid; FA—fumaric acid; BA—benzoic acid; CF—calcium formate; FO—formic acid; CA—citric acid; AA—acetic acid. Matrix coated OA blend—17% fumaric acid, 13% citric acid, 10% malic acid, and 1.2% MCFA (capric and caprylic acid) and carrier. ADG—average daily gain; BWG—body weight gain; ADFI—average daily feed intake; FCE—feed convection efficiency.

**Table 3 animals-14-00762-t003:** Application of commercial and non-commercial essential oils in pig nutrition and health.

Name	Components	Dietary Dose and Duration	Main Findings	References
Delacon blend	40% Common Fenugreek seed, 12.5% Subterranean Clove, 7.5% Cinnamomum Cassia Presl, and 40% Kaolin (2SiO_2_.Al_2_O_3_.2H_2_O)	0.04%. 42–60 days	Improved growth performance, apparent ileal digestibility	[37]
ORSENTIAL	1.1% Thymol + 2.2% Carvacrol	300–1000 g/tonne. 54 days	Higher ADG, lower incidences of diarrhea, reduced fecal ammonia emissions and blood urea nitrogen, increased serum IgG	[38]
ColiFit Icaps C	Trans-cinnamal Ehyde, eugenol, carvacrol, thymol, and diallyl disulfure at 101,218; 12,400; 6514; 4359 and 1123 mg/kg, respectively.	1 kg/tonne, 7 days	Higher fecal lactobacilli, increased lactobacilli/coliform ratio against enterotoxigenic *Escherichia coli* (ETEC) F4 strain COLI30/14-3	[39]
PEP1000-1, Biomin Inc.	Anis oil, citrus oil, oregano oil	0.1%	Improved diarrhea score	[27]
Next enhance 150, NE150	thymol 25% and carvacol 25%		Improved nutrient digestibility, antioxidant ability, intestinal morphology, and digestive enzymes in weaned pigs	[26]
*Essential oil* blend	*Cinnamomum zeylanicum* and *Trachyspermum capticum*	0.3 and 0.4 g/kg, resp duration: 63 days	Increased HDL concentration at day 28 Increase ImmunoglobulinM from day 28–56 Serum pro-inflamattory cytokines (IL-6) decreased from day 28–56, higher lactobacilli and lower fecal enterobacterial populations	[40]

ADG—average daily weight gain; HDL—high density lipoprotein.

**Table 5 animals-14-00762-t005:** Effects of organic acids and essential oils on voluntary feed intake.

Feed Additive	Pig Group	Effect on VFI	References
EO	Piglets	Increase	[26]
EO	Piglets	Decrease	[46]
EO	Piglets	Decrease	[47]
EO	Piglets	Increase	[6]
EO blend	Piglets	NS	[48]
OA	Piglets	NS	[49]
OA	Finishing pigs	Increase	[49]
OA blend	Weaned piglets	NS	[50]
EO + OA	Nursary piglets	Increase	[26]
EO + OA	Finishing pigs	NS	[51]

VFI—voluntary feed intake; EO—essential oils; OA—organic acids; NS—not significant.

**Table 6 animals-14-00762-t006:** Effects of organic acid and essential oil blends on pig performance in relation to age.

OA and EO Blends	Target	Dose	Results	References
* BA + EO	Weanling pigs	3.0 and 0.1%	No effect on growth performance, metabolites, cytokines, intestinal microbiota	[81]
OA (FA, AA, CA, PA, and Ca) + EO (thyme, nettle, oak, and balm)	Weaned piglets	CR + 0.5% EA + 0.3% OA	Improved daily gains at later growth stage, higher protein quality, 6.9% cholesterol reduction	[82]
BA + EO (thymol, 2-methoxyphenol, eugenol, piperine, and curcumin)	Weanling pigs	2/3/4 g/kg at 1.8 BA + 0.072 EO, 1.8 BA + 0.072 EO and 1.8 BA + 0.072 EO levels	Increased net revenue when BA + EO at 3 or 4 g/kg	[83]
FA + FormaXOL^TM^	Finisher pigs	4 kg/tonne	Reduces Salmonella shedding and seroprevalence at longer supplementation duration but increased feed cost per live weight gain	[84]
BA 50%, Calcium formate 3% and FA 1% + Thymol 25% and carvacrol 25%	Weaned piglets	1.5 g/kg OA + 30 mg/kg EO	Complementary effect on growth performance, little interactive effects on intestinal health between EO and OA	[26]
Cinnamaldehyde 15%, thymol 5%, CA 10%, SA 10%, MA 6.5% and FA 13.5%	Weaned piglets	1 kg/tonne	Improved growth performance and fecal microbes, modulate serum immune parameters, increased isovaleric acid	[85]
PEP1000-1^®^, (anis, citrus, oregano oils, and natural Flavors) + Biotronic^®^, (PA and LA)	Nursery piglets	0.4% and 0.2%, resp.	Matched growth performance of antibiotic supplement	[40]
Bamboo Vinegar + Acidifier I Bamboo Vinegar + Acidifier II	Weaned piglets	0.4% BV + 0.25% Acidifier	Wider species richness and bacterial community diversities in feces	[81]

CR—complete ration; PA—phosphoric acid, LA—lactic acid; FA—fumaric acid; BA—benzoic acid; CF—calcium formate; FO—formic acid; CA—Citric acid; AA—acetic acid; MA—malic acid; SA—sorbic acid. Acidifier I—LA, CA, MA, TA, and PA mixed at 20:20:10:15:35; Acidifier II—LA, CA, MA, TA, and PA mixed at 40:20:20:20:0; * FormaXOL^TM^—encapsulated blend of formic acid, citric acid, and essential oils from citrus fruit extract, cinnamon, oregano, thyme, and capsicum, Kemin Industries, Inc., Southport, UK, UK*EO—essential oils (CRENA; DSM Nutritional Products, LLC, Belvidere, IL, USA); BA—benzoic acid ((Vevovitall^®^, DSM Nutritional Products Inc., Parsippany, NJ, USA).

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
