# Peer review of "Potential of Organic Acids, Essential Oils and Their Blends in Pig Diets as Alternatives to Antibiotic Growth Promoters"

_animals, 2024, doi:10.3390/ani14050762_

Round 1
Reviewer 1 Report
Comments and Suggestions for Authors
Manuscript animals-2760774, entitled “Potential of organic acids, essential oils and their blends in pig diets as alternatives to antibiotic growth promotors”
This review article provides useful information on the utilization of organic acids, essential oils and their blends as alternatives to antibiotic growth promoters in pigs. However, there are some points that should be corrected or clarified.
· In abstract and simple summary, please add a concluding sentence. Finally, organic acids, essential oils and their blends have positive or negative effects?
· When study is initially used in text, dose and duration of supplementation should be provided.
· Please correct the numbering of sections and subsections. You should better organize the article
· According to a quick search in databases, there are several studies that describe the effects of EOs in pigs. Please try including the majority of them in your review article. At the same time, please use up-to-date literature (after 2020)
· Please provide a title for Figure 5
· Throughout the text, you can use OAs and EOs instead of organic acids and essential oils, respectively
L3: “promoters”
L13: “possess” instead of “have”
L14: Please delete “and an effect on pork quality”
L18: “Application” instead of “Development”
L19: “has been increased”
L20: What do you mean by “to prevent diet associated stress”?
L27: “appeared” instead of “have been said”
L30: “examine” instead of “deliberate on”
L36: “Over the years, there has been a growing global interest in the…”
L40: What do you mean by “to prevent diet-associated stress”?
L38-47: This part is repeated as in abstract. Please rephrase
L48: Please delete “on”
L50: “reported” instead of “said”
L54-55: “…stability. As a result, an improved pig performance comparable to antibiotic growth promoters is observed [6, 7, 8]. The focus of this review is to provide a…”
L70-71: “…microorganisms [9]. However, their chemical composition and bacterial spectrum vary widely.”
L74-75: Please rephrase
L76-77: Please delete (repetition; L67-68)
L86: Different section. This is not a part of “Mode of action of Antibiotic Growth Promoters”
L104-105: Please rephrase
L119: “…grew by 6.6% from…”
L123-124: Please rephrase
L129: “is” or “are” expected?
Table 2: What is the meaning of “Acid 2” in 4th row?
L183: “provide” instead of “give”
L190: “In vitro antibacterial properties”
L203: “It leads…”?
L209: “This will result…”?
L223: Something is missing. The sentence is incomplete
L236: “Feed intake change…” Reduction or increase?
L238 and throughout the text (L333, 335, 343, 350, 351, 355 etc): Please add the name of authors before “[ ]”, as “Lan et al. [36] reported a range…”
L260-261: Essential oils improved apparent…”
L274: Please rephrase
L280: “Protective acids”?
L285-296: You refer to microbiota also in the previous section
L315-316: “Supplementation” instead of “Supplementing”
L321: “Oxidative stress is commenced when the amount…”
L334-335: “…that carvacrol added in drinking water reduced the level of DNA lesions induced in freshly isolates hepatocytes and testicular cells in pigs.”
L356: “…no effect on meat marbling, colour, cooking…”
L358: “…do not alter carcass composition.”
L367: “olfactory effects”?
L393: “comparably”?
Comments on the Quality of English Language
Extensive editing of English language required
Reviewer 2 Report
Comments and Suggestions for Authors
I am pleased that I could review this comprehensive article,
The topic is a very interesting and extremely important issue in pig production.
The study was prepared at a high substantive level.
I only have small reservations regarding the description of the impact of the quality of essential oils on the quality of meat.
It is written several times that changing the diet does not affect the quality of the carcass and meat.
This is an incorrect statement or it simply does not specify what kind of diet change.
Because nutritional modifications are the best way to modify the quality of pork. There are many scientific studies confirming that the addition of oils, herbs or new plants in the diet of fattening pigs has a positive effect on the quality of pork: e.g.
https://sciendo.com/pl/article/10.1515/aoas-2016-0002
Https://www.mdpi.com/2076-2615/12/12/1526
I think it's important to mentionted that scientific research proves that herbs and the essential oils and biologically active substances they contain influence the quality of pork, but there is not enoughtoo research confirming the influence of essential oils themselves on the quality of pork.
Most of my comments concern technical errors.
Particular attention should be paid to adding connections between figures and tables and the text.
I have presented detailed comments below:
1. In my opinion it is better to add: “ Scientific research shows that Alternative growth pro-moters such as….” ( line 26)
2. Better and more accurate is the objective of the review written in the "introduction" section (line 29)
3. In my opinion it is better to write: “ …. Prevents the effects of stress resulting from changing feed in the diet….” ( line 38-41)
4. “they also improve mineral utilization, act as an energy source, promote endogenous enzyme secretion, and improve immunity and antioxidant stability.” - references should be added (line 54)
5. Reword two sentences. First, why diarrhea after weaning? Secondly, what does a small amount of hydrochloric acid lead to (line 63-67)
6. Add references confirming growth performance (line 68)
7. “In growing pigs’ antibiotics improve the growth performance and feed efficiency of growing pigs”- delete, this sentence is already above (line 76-77)
8. “Huge risk to the environmental”- in my opinion this is better ( line 85)
9. "organic acids are categorized into three main functional classes" - list these groups in the text, mark the first, the second and the third (line 90- 99)
10. General structures of a. - add brackets (above line 100)
11. "structures" - move above the drawing (line 100)
12. Place a reference (connector) to the figure 1 in the text (line 101)
13. Figure 1 without special characters. Apply to the entire article (line 101)
14. Expand abbreviation CARG ( line 123)
15. I'm not sure if this bracket should be here or just in the references (line 124)
16. Explain the abbreviation the first time you use it (line 135)
17. Remove the dot after the bracket (line 139)
18. Too long a space between the dot and the word "table 2" (line 154)
19. Add explanation of abbreviations; BWG, ADFI, FCE (line 158)
20. Remove special characters - space symbol and enter. All figures in the article without special characters (line 176)
21. Write "dose" with a lower case letter, the same in table 2 (line 189)
22. Add explanation of abbreviations bellow the table 3: ADG, HDL ( above line 190)
23. Add a reference to sentence “ Esential oils….”( 193)
24. Remove special characters - space symbol and enter. All figures in the article without special characters (line 228)
25. It seems to me that [50] is written in a different font(line 228)
26. Place a reference (connector) to the drawing in the text (line 229)
27. In my opinion, it is better to use an intense smell instead of a pungent smell (line 233)
28. In my opinion, it is better : “Lan, et al., reported a range of 3%-19% increase in feed intake [36].” (line 238)
29. In my opinion, instead of "age", it is better to use "pig technological group" or "pig group" and instead of piglets, "nursery piglets". Nomenclature compatible with table 6. (line 246)
30. Add a dot after [34] (line 251)
31. Change the order of these two sentences (line 269- 272)
32. In my opinion, it is better: “Hhowever, in a study by Kong et al., butyric acid did not affect the histology of growing-finishing pigs, only mucosal depth was larger and this can be attributed to better gut integrity in older animals [68]. ( line 306- 308)
33. Rephrase in a similar way to the point: 28 i 32 ( line 317- 318)
34. Expand the abbreviation when first using it (line 321)
35. Rephrase in a similar way to the point: 28 i 32 (line 333)
36. Rephrase in a similar way to the point: 28 i 32 (line 335)
37. Rephrase in a similar way to the point: 28 i 32 (line 350)
38. Rephrase in a similar way to the point: 28 i 32 (line 351)
39. To clarify - in my opinion, the text concerns not the physicochemical composition of the meat itself, but the physical parameters and technological and consumer quality (or simply the broadly understood quality of pork) (line 353)
40. Specify what changes in the diet. Because generally nutritional modifications are the best way to modify the quality of pork. There are many scientific studies confirming that the addition of oils, herbs or new plants in the diet of fattening pigs has a positive effect on the quality of pork: e.g.
Https://www.mdpi.com/2076-2615/12/12/1526
Https://sciendo.com/pl/article/10.1515/aoas-2016-0002
(line 354- 355).
41. “" diet alters carcass composition" - it's unclear. What exactly it changes? ( line 355)
42. Not in the finisher pig but in the meat obtained from finisher pigs ( line line 355-357)
43. “)rganic acids are considered growth promoters and do not alter the composition of a carcass” - there is no certainty that they will not change, it is safer to write that there are no scientific studies confirming that.... (line 357-358)
44. Clarify what the effect was ( line 364)
45. In my opinion it should be figure 4 - numbering error (line 378)
46. I understand that this change is drastic but beneficial - it would be good to clarify ( line 386)
47. No signature under the figure. Remove characteristic marks from the figure. The graphic quality of the figure is not very good (line 387)
48. Too large a gap (line 517)
49. Lost word “Yan”
50. Too large a gap (561)
51. Lack of justification... (562-568)
52. Too large a gap (575)
53. Too large a gap (579)
54. Too large a gap (583)
55. Too large a gap (601)
56. Too large a gap (607)
57. Too large a gap (615)
58. Include publications in references in accordance with the editor's requirements (currently there is one full form and one short form) (line 400- 638).
Round 2
Reviewer 1 Report
Comments and Suggestions for Authors
Authors made the majority of the necessary amenments. Howerer, some minor corrections should be further carried out:
L14: The word "contain" is not correct here
L15: "look at"?
L23: "has increased"?
In page 3 please provide a title for Figure showing the structure of organic acids
L196: "are not compounding"?
The quality if Figures 1, 2 and 3 is not the appropriate
Table 5: "Nursery piglets"
Please check numbering; for example, L99: 3.1, L150: 3.2, L157: 3.2.1., L176: 3.2.2., L184: L3.2.3. - L195: 4.1, L208: 4.2, L224: 4.3, L238: 4.4
L400: "olfactory effects"?
Comments on the Quality of English Language
Moderate editing of English language required
